# Constructing of Ni-Nx Active Sites in Self-Supported Ni Single-Atom Catalysts for Efficient Reduction of CO_2_ to CO

**DOI:** 10.3390/nano15060473

**Published:** 2025-03-20

**Authors:** Xuemei Zhou, Chunxia Meng, Wanqiang Yu, Yijie Wang, Luyun Cui, Tong Li, Jingang Wang

**Affiliations:** Institute for Advanced Interdisciplinary Research (iAIR), School of Chemistry and Chemical Engineering, University of Jinan, Jinan 250022, China; zhouxuem2025@163.com (X.Z.); mengcunxia123@163.com (C.M.); 202211100020@stu.ujn.edu.cn (W.Y.); wangyj1004@163.com (Y.W.); serenacly@163.com (L.C.); 15045328637@163.com (T.L.)

**Keywords:** CO_2_ electroreduction reaction, Ni-N-C electrocatalyst, single atom, electrospinning technology

## Abstract

The electrochemical carbon dioxide reduction reaction (CO_2_RR) represents a promising approach for achieving CO_2_ resource utilization. Carbon-based materials featuring single-atom transition metal-nitrogen coordination (M-N_x_) have attracted considerable research attention due to their ability to maximize catalytic efficiency while minimizing metal atom usage. However, conventional synthesis methods often encounter challenges with metal particle agglomeration. In this study, we developed a Ni-doped polyvinylidene fluoride (PVDF) fiber membrane via electrospinning, subsequently transformed into a nitrogen-doped three-dimensional self-supporting single-atom Ni catalyst (Ni-N-CF) through controlled carbonization. PVDF was partially defluorinated and crosslinked, and the single carbon chain is changed into a reticulated structure, which ensured that the structure did not collapse during carbonization and effectively solved the problem of runaway M-Nx composite in the high-temperature pyrolysis process. Grounded in X-ray photoelectron spectroscopy (XPS) and X-ray absorption fine structure (XAFS), nitrogen coordinates with nickel atoms to form a Ni-N structure, which keeps nickel in a low oxidation state, thereby facilitating CO_2_RR. When applied to CO_2_RR, the Ni-N-CF catalyst demonstrated exceptional CO selectivity with a Faradaic efficiency (FE) of 92%. The unique self-supporting architecture effectively addressed traditional electrode instability issues caused by catalyst detachment. These results indicate that by tuning the local coordination structure of atomically dispersed Ni, the original inert reaction sites can be activated into efficient catalytic centers. This work can provide a new strategy for designing high-performance single-atom catalysts and structurally stable electrodes.

## 1. Introduction

The rapid industrialization and extensive consumption of fossil fuels have driven a sustained increase in atmospheric CO_2_ emissions, posing global challenges such as climate change, sea-level elevation, ocean acidification [1,2,3]. To realize carbon peaking and carbon neutrality objectives, it is crucial to develop strategies that simultaneously mitigate atmospheric CO_2_ levels and convert this greenhouse gas into value-added products [4,5]. Electrochemical CO_2_ reduction reaction (CO_2_RR) technology has emerged as a promising solution, enabling the transformation of CO_2_ into industrially relevant chemicals and fuels (e.g., CO, CH_3_OH, C_2_H_4_) through sustainable pathways [6]. Notably, CO stands out as a versatile chemical feedstock for synthesizing essential industrial compounds including methanol and phosgene [7], making its production via CO_2_RR particularly valuable for carbon resource utilization. The CO_2_RR process involves a complex multi-step mechanism where proton-coupled electron transfer (PCET) constitutes the rate-determining step [8]. However, the inherent stability of CO_2_ molecules, primarily due to their high C=O bond energy (~806 kJ·mol^−1^) [9], imposes substantial activation barriers that necessitate high overpotentials. This thermodynamic limitation not only leads to excessive energy consumption but also intensifies competing side reactions, particularly the hydrogen evolution reaction (HER) [10]. Therefore, the rational design of electrocatalysts that can effectively lower activation potentials while maintaining high product selectivity and catalytic efficiency represents a critical scientific challenge that demands immediate attention [11].

The catalysts for the reduction of CO_2_ to CO included metal-based catalysts [12], alloy catalysts [13], metal oxides [14], carbon-based catalysts (doped carbon materials) [15], single-atom catalysts (SACs) [16,17]. Precious metal catalysts (notably Au [18], Ag [19], and Pd [20]) demonstrate superior performance owing to their distinctive electronic configurations, optimized adsorption energetics, and rapid charge transfer kinetics. Nevertheless, practical deployment of these catalysts faces three fundamental constraints: prohibitive material costs, limited natural abundance, and insufficient mechanistic understanding of active sites. Carbon-supported nonprecious metal catalysts (e.g., Ni [21], Fe [22]) present cost advantages but confront intrinsic electronic structure limitations. The mismatch between their Fermi levels/electron density distributions and the energy requirements for optimal CO_2_ adsorption results in unstable adsorbate configurations and elevated activation barriers [23]. This electronic incompatibility, coupled with insufficient CO_2_ binding strength, leads to inefficient molecular capture and HER dominance under low overpotentials [24]. Additional challenges include maintaining homogeneous active site distribution, as aggregation phenomena during synthesis cause site blocking and performance degradation [25]. The emergence of single-atom catalysis (SACs) since 2011 has fundamentally transformed CO_2_RR through dual synergistic mechanisms [26]: (1) Electronic modulation via metal-support interactions (e.g., Fermi level adjustment through charge transfer) enhances d-electron density near the Fermi level, effectively lowering CO_2_ activation barriers; (2) Atomic-level coordination engineering (e.g., tailoring N coordination numbers in M-N_x_ configurations) enables precise control over intermediate stabilization (e.g., *COOH) while suppressing HER. These characteristics, combined with near-theoretical atomic utilization, allow SACs to overcome the efficiency-selectivity paradox inherent to conventional catalysts. Particularly noteworthy are nitrogen-doped carbon matrices that simultaneously stabilize single metal atoms through robust M-N bonds and provide secondary active sites (e.g., pyridinic N) to synergistically enhance CO_2_ adsorption and proton transfer [27]. Among SAC configurations, Ni-based systems have emerged as frontrunners due to their optimal d-band positioning for intermediate binding [28]. The Ni-N_4_-C configuration is prone to form *COOH due to the presence of high free energy barriers [29,30]. However, the central metal atom geometry and electronic configuration can be changed by strategies such as screening the substrate and changing the coordination environment [31,32,33]. Axial modification of the Ni-N_4_ active center by heteroatom (N, P, B, and other atoms) doping method can promote CO_2_ activation, lower the free energy barrier required for the formation of *COOH, and promote the dissociation of CO, thus improving the kinetics and catalytic activity [34,35,36,37]. For example, Zhang et al. developed a new catalyst Ni6@Ni-N_3_ to unsaturated Ni-N_3_ sites by integrating Ni nanoclusters. Ni_6_@Ni-N_3_ exhibited up to 99.7% CO Faraday efficiency at 500 mA cm^−2^ under −1.15 V vs. RHE [38]. For instance, Zhao et al. developed MOF-derived Ni SACs with engineered coordination environments, which the faraday efficiency of CO could achieve 95.6% at −0.65 V vs. RHE [39]. Current research focuses on overcoming two critical synthesis challenges: (i) preventing metal aggregation during precursor reduction and pyrolysis, and (ii) developing self-supporting architectures. The fiber membrane treated with defluorination, and cross-linking operations does not collapse after carbonization and can maintain its structural integrity. Therefore, it can effectively solve the problem of uncontrolled M-Nx compounding in high-temperature pyrolysis [40,41]. Metal-doped EFCMs processed through nitrogenation and carbonization form binder-free electrodes that eliminate adhesive-induced site blocking, thereby enhancing both catalytic activity and operational stability [42].

In this study, electrospinning technology was used to fabricate a PVDF-Ni composite precursor with a high specific surface area and a nanoporous structure. The precursor was subjected to high-temperature carbonization and nitrogen doping, successfully resulting in the formation of a Ni-N_4_-C structure-based single-atom catalyst. During the carbonization process, defluorination and cross-linking of the PVDF fibers preserved the original three-dimensional network structure, allowing the material to be directly utilized as a self-supporting electrode. This eliminated the instability issues often associated with the detachment of powder catalysts. The self-supported catalyst demonstrated excellent selectivity in the electrochemical reduction of CO_2_ to CO.

## 2. Materials and Methods

### 2.1. Materials

Polyvinylidene difluoride (99%, (CH_2_CF_2_) n) were purchased from Solvay. Nickel chloride hexahydrate (99.9%, NiCl_2_·6H_2_O), N,N-dimethylformamide (99.9%, DMF), nano magnesium oxide (99.9%, MgO), acetone (99.9%, CH_3_COCH_3_), p-xylene diamine (99%, C_6_H_4_(CH_2_NH_2_)_2_) were from Shanghai McLean Reagent Co., Ltd. Argon (99.7%, Ar), carbon dioxide (99.9%, CO_2_) and ammonia (99.9%, NH_3_) were from Jinan Yao tian Instrument Jinan., China.

### 2.2. Preparation of Catalysts

Three types of catalysts, namely Ni-doped PVDF catalyst carbonized in an argon-ammonia blended atmosphere (Ni-N-C), Ni-doped PVDF catalyst carbonized in an argon environment (Ni-C), and PVDF carbonized under an ammonia atmosphere (N-C), were synthesized. Taking Ni-N-CF as an example: 1.2 g of polyvinylidene fluoride tetrachloroethylene (PVDF) was mixed thoroughly with 4 mL of acetone and 6 mL of N, N dimethylformamide (DMF). An amount of 0.0012 g of nickel (II) chloride hexahydrate (NiCl_2_·6H_2_O) was added into it. Throughout 12 h of magnetic agitation at 25 °C, the precursor mixture gradually transformed into a viscously uniform system suitable for subsequent spinning processes. The spinning solution was loaded into a 10 mL medical syringe and electrospun under a high-voltage electric field of 20 kV. The process parameters were set as follows: needle-to-collector distance was 15 cm, collector rotation speed was 80 rpm, and peristaltic pump flow rate was 1.02 mL h^−1^. The resulting fibrous membrane, denoted as Ni-PVDF, was vacuum-dried at 80 °C for 12 h and subsequently densified using a hot press (2 MPa, 120 °C). A cross-linking solution was prepared by mixing 15 mL of methanol, 1 g of NaOH, 4 g of 1,4-phenylenediamine, and 2 g of MgO nanoparticles. The Ni-PVDF membrane underwent sequential treatments: (1) immersion in 1 M HNO_3_ for 1 h; (2) rinsing with deionized water for 6 h; (3) ultrasonic cleaning in methanol and n-hexane for 2 h each. This cleaning cycle was repeated three times, followed by vacuum drying at 60 °C. Controlled carbonization was performed in a tube furnace under an Ar/NH_3_ mixed gas flow (1:2). The temperature was ramped at 5 °C min^−1^ to 1000 °C and maintained for 2 h, yielding nitrogen-doped nickel-carbon composites (denoted as Ni-N-CF). Control samples were prepared under modified conditions: (1) N-CF synthesized without nickel precursors; (2) Ni-CF obtained via carbonization in pure Ar atmosphere.

### 2.3. CO_2_ Reduction Reaction Measurements

All electrochemical tests involved in this experiment were carried out using a traditional three-electrode system in CO_2_ saturated NaHCO_3_ solution (0.5 mol/L) using a closed classical H electrolyzer equipped with an electrochemical workstation (CHI660D, Shanghai Chenhua Instrument Shanghai, China). The prepared catalyst carbon fiber membrane was cut into a rectangle of 2 cm × 1 cm as the working electrode (to ensure that its area immersed in the electrolyte was 1 cm × 1 cm), the reference electrode was a saturated calomel electrode (SCE), and the Pt piece was the counter electrode. The left and right chambers were filled with 15 mL of CO_2_-saturated NaHCO_3_ solution (0.5 mol/L) during testing. Prior to testing the CO_2_RR performance, a 30 min purge was performed with a 30-sccm flow rate of Ar to test the LSV, followed by a 30 min purge with a 30-sccm flow rate of CO_2_ to test the LSV. The two LSV curves were compared to derive a range of potentials for the test. A potential was applied to the catalyst during the test using an electrochemical workstation, and after 1 h of reaction, 10 mL of gas was removed using a syringe and kept sealed to test the gas phase product. The liquid phase products were tested by removing 10mL of liquid phase products. The gas phase and liquid phase data were tested at different voltages. All applied potentials were converted into reversible hydrogen electrode (RHE) using the following equation: E (vs. RHE) = E (vs. SCE) + 0.24 + 0.059 × pH. The gas-phase products were analyzed by online gas chromatography (GC, HF-901, Shandong Huifen Instrument Jinan, China) equipped with a thermal conductivity detector (TCD) and a flame Ionization detector (FID). No liquid products were detected by nuclear magnetic resonance (NMR) spectrometer.

## 3. Results and Discussion

The synthesis procedure of the Ni-based N-doped carbon fiber catalyst (Ni-N-CF) is shown in Figure 1a. The preparation process of the catalyst in this study mainly involved three key steps. First, the precursor was prepared using electrospinning technology. Second, the precursor underwent defluorination and cross-linking to form a stable carbon skeleton structure. Finally, the target sample was obtained by nitrogen-doped carbonization at a high temperature of 1000 °C. The specific preparation procedure is described in detail in the Appendix A. Figure 1b,e are digital photographs of the Ni-PVDF fiber membrane before and after carbonization. It can be observed that Ni-N-CF did not exhibit significant shrinkage after calcination. A cross-sectional image was obtained using SEM, measuring the average thickness of the electrode to be 180 ± 5 μm (Appendix A). The electrode has a single-layer structure, but its interior consists of numerous intersecting fibers, with no additional functional layers stacked on top. SEM results demonstrate that the internal structure of both Ni-PVDF (Figure 1c,d) and Ni-N-CF (Figure 1f,g) fiber membranes maintained a similar three-dimensional mesh structure before and after carbonization. Infrared evidence (Appendix A) showed that the double bonds of different carbon chains within PVDF underwent cross-linking to form a network structure [43,44]. The surface of the Ni-PVDF fibers was relatively smooth, while after carbonization, the fiber surface exhibited noticeable pits and irregular rough structures. Fiber diameter analysis revealed that the average fiber diameter before carbonization was approximately 0.72 μm (Appendix A), and after carbonization, it increased slightly to 0.76 μm (Appendix A). These results indicated that a three-dimensional network self-supporting catalyst with a stable structure has been prepared.

Structural characterization of the catalyst was conducted using TEM, SAED, EDS, and nitrogen adsorption–desorption analysis. As shown in Figure 2a, the Ni-N-CF catalyst exhibits a uniform internal architecture with distinct brightness variations, revealing a well-developed porous structure. Contact angle measurements (Appendix A) demonstrate significantly enhanced hydrophilicity (contact angle reduced from 115.2° to 11.92°) after nitrogen doping and carbonization, which facilitates CO_2_ adsorption/desorption processes. Atomic-scale imaging in Figure 2b reveals a characteristic 0.34 nm lattice spacing corresponding to the (002) plane of graphitic carbon, with no observable metallic Ni crystalline phases, consistent with the diffuse SAED pattern (inset). As shown in Appendix A, the XRD patterns exhibit two characteristic diffraction peaks at 23° and 40°, corresponding to the (002) and (111) crystallographic planes of amorphous carbon, respectively. Elemental mapping in Figure 2c confirms the homogeneous distribution of C, N, O, and Ni throughout the carbon nanofibers, indicating excellent metal dispersion. Nitrogen sorption analysis reveals a type IV isotherm with H_3_ hysteresis (Figure 2d), characteristic of mesoporous materials, yielding a remarkable specific surface area of 606.83 m^2^·g^−1^ for Ni-N-CF-73-fold higher than uncarbonized Ni-PVDF (8.36 m^2^·g^−1^). The corresponding pore size distribution (Figure 2e) confirms a hierarchical pore structure spanning micro- (<2 nm), meso- (2–50 nm), and macropores (>50 nm), with a total pore volume of 0.55 cm^3^·g^−1^. This three-dimensional interconnected porosity not only provides abundant active sites through structural defects created during N-doping but also establishes efficient transport pathways for CO_2_ molecules and electrolyte ions [45,46].

The phase composition and structure of the Ni-N-CF, Ni-CF, and N-CF catalysts were systematically characterized using HAADF-STEM, Raman spectroscopy, XPS, and XAFS. Figure 3a,b shows HAADF-STEM images at different magnifications. No large Ni nanoparticles were observed; the bright spots (regions indicated by red circles) in the images correspond to Ni single atoms [47]. Figure 3c displays the Raman spectra featuring characteristic D (1340 cm^−1^) and G (1590 cm^−1^) bands [48]. The progressive increase in ID/IG ratio from 2.35 (N-CF) to 2.53 (Ni-N-CF) with simultaneous Ni/N doping suggests enhanced graphitic ordering [49]. Furthermore, a 5 cm^−1^ positive shift in the G-band position observed for Ni-N-CF compared to Ni-CF indicates nitrogen-induced lattice distortion, which generates structural defects exposing additional active sites for catalytic reactions [50]. To further investigate the relationship between the electronic structure of the catalyst and its catalytic activity, XPS analysis was performed. The results in Appendix A revealed that Ni-N-CF contained only nitrogen (N), carbon (C), and oxygen (O); Ni-CF exhibited carbon and oxygen, while N-CF showed the presence of nitrogen and oxygen. Figure 3d showed the high-resolution N1s spectrum of the Ni-N-CF catalyst. Compared to the N1s spectrum of N-C shown in Appendix A, a distinct Ni-N bond peak was observed at 399.5 eV, indicating that Ni metal atoms form a stable Ni-N structure within the carbon fiber framework through coordination with N atoms. Additionally, peaks for pyridinic nitrogen (398.3 eV), pyrrolic nitrogen (400.8 eV), graphitic nitrogen (402.1 eV), and oxidized nitrogen (403.8 eV) were also observed, confirming the presence of various nitrogen doping types in the catalyst [51,52]. In particular, the high content of pyridinic nitrogen was likely to facilitate the activation of CO_2_ molecules, thereby promoting the catalytic reaction [53,54,55]. The content of pyridinic nitrogen in the Ni-N-CF catalyst was significantly higher than that in the Ni-CF catalyst (Appendix A). Further analysis of the high-energy-resolution O 1s spectrum of the Ni-CF catalyst demonstrated the existence of Ni-O bonds (Appendix A). The XAFS spectrum of the Ni K-edge was characterized (as shown in Figure 3e) to gain deeper insights into the chemical state and coordination environment of Ni. An analysis of this figure revealed a slight right shift for Ni-N-CF, indicating that the average oxidation state of Ni atoms in the catalyst was above zero. As shown in Figure 3e, the Ni K-edge XANES spectra show that the near-edge absorption energy of Ni-N-C was located between those of the Ni foil and NiO, indicating that the average valence state of Ni in Ni-N-C was between 0 and +2. The fingerprint peak was around 8336 eV, which was assigned to the 1s-4p_z_ transition of the square planar Ni-N_4_ moiety [56]. Figure 3f displayed the Fourier transform of the EXAFS spectrum with k^3^-weighted χ(k) function, where only one peak at 1.39 Å was observed in Ni-N-CF, corresponding to the Ni-N bond [16]. This indicated that Ni atoms were isolated and dispersed on the nitrogen-doped carbon support. The ICP-MS analysis (Appendix A) indicated that the Ni content in Ni-N-CF was 0.08 wt%. The above results demonstrate the successful preparation of the Ni-N-CF single-atom catalyst.

To investigate the catalytic activity of the Ni-N-CF catalyst designed, particularly the advantages of the N-doped mesh self-supporting structure, we conducted CO_2_RR tests using a three-electrode system in an H-type electrolytic cell. All applied potentials were referenced to the reversible hydrogen electrode (RHE) in this work. As the LSV curve in Figure 4a, the Ni-N-CF catalyst showed a more significant increase in the current density for CO_2_ in CO_2_-saturated NaHCO_3_, which was attributed to the CO_2_ reduction. At −0.65 V vs. RHE, Ni-N-CF exhibited a current density of 2.8 mA cm^−2^, surpassing N-C (0.1 mA cm^−2^) and Ni-CF (1.5 mA cm^−2^). The activity and selectivity of the Ni-N-CF catalyst were tested after one hour of electrolysis at a constant potential. Only CO and H_2_ were detected over the whole investigated potential range. The maximum FE of CO reached 92% at −0.65 V vs. RHE (Figure 4b), which the selectivity of Ni-N-CF for reduction to CO was much higher than N-C (1%) and Ni-CF (3%) (Figure 4c). In addition, Figure 4d showed that the current density for CO_2_RR of Ni-N-CF was superior to that of other catalysts at all tested potentials. The results obtained from XPS, aberration-corrected microscopy, XANES, and electroreduction data indicated that nitrogen doping keeps nickel in a low oxidation state, and Ni-N was the main active site for CO_2_RR in Ni-N-CF. This confirms the significant role of nitrogen doping in enhancing the performance of Ni-CF, thereby strengthening the argument for the synergistic mechanism between nickel single atoms and nitrogen doping. As shown in Figure 4e, during the continuous 12 h CO_2_ reduction to CO reaction at -0.65 V vs. RHE, the Ni-N-CF catalyst exhibits no significant decay in FE and current density, indicating its robust stability. In addition, compared to other recently reported Ni-N_4_ catalysts, Ni-N-CF seems to exhibit superior selectivity for CO products in CO_2_ RR [39,57,58,59,60,61,62,63] (Appendix A). These results indicated that Ni-N_4_-C species possess excellent activity and selectivity, with Ni-N being the decisive catalytic site in the carbon dioxide reduction process. This is consistent with reported findings in the literature on M-N-C catalysts [64,65].

## 4. Conclusions

In summary, we propose a rational design and simple synthesis method to construct an atomically dispersed nitrogen-doped three-dimensional network carbon nanofiber membrane (Ni-N-CF) electrocatalyst. The network structure formed after the cross-linking of PVDF fiber membrane can inhibit the structural collapse during carbonization. The Ni single atoms and nitrogen doping form Ni-N coordination on the self-supported porous fiber membrane, which is beneficial for the reduction of CO_2_ to CO. The Ni-N-CF structure exhibits excellent selectivity for CO production (with a Faradaic efficiency of 92%) and long-term stability for CO_2_RR at a low overpotential (−0.65 V RHE). XPS, XAFS, and electrochemical performance analyses confirm that Ni-N is the main active site of Ni-N-CF. This study provides a practical and efficient approach for developing atomically scaled three-dimensional electrocatalysts.

## Figures and Tables

**Figure 1 nanomaterials-15-00473-f001:**
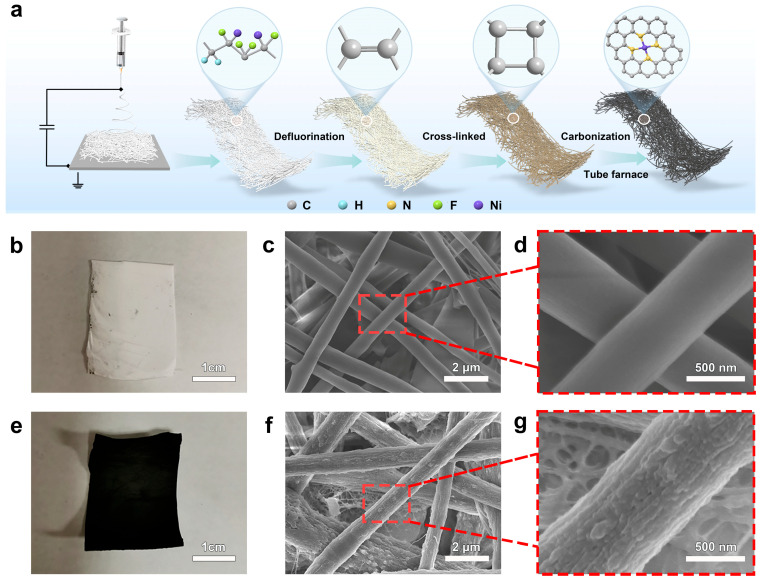
(**a**) Schematic diagram of the formation of Ni-N-CF (**b**) Digital image of Ni-PVDF before carbonization. (**c**,**d**) SEM images of Ni-PVDF at different magnifications before carbonization. (**e**) Digital image of Ni-N-CF after carbonization. (**f**,**g**) SEM images of Ni-N-CF at different magnifications after carbonization.

**Figure 2 nanomaterials-15-00473-f002:**
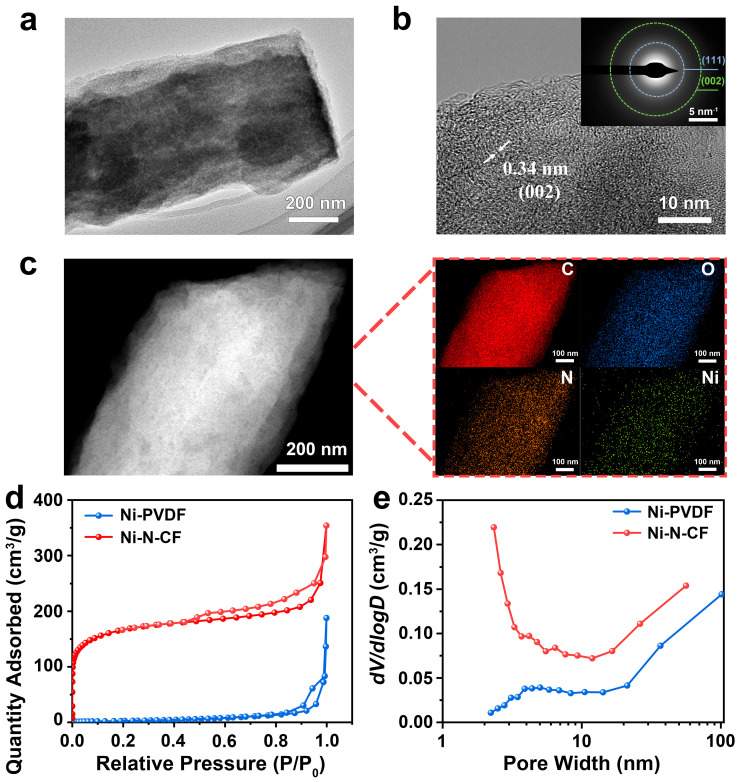
(**a**) TEM image of Ni-N-CF. (**b**) HRTEM image (inset is the SAED image) of Ni-N-CF. (**c**) Elemental mapping analysis of Ni-N-CF and the distributions of C, N, Ni, and O elements. (**d**) N_2_ adsorption–desorption isotherm of Ni-N-CF and Ni-PVDF catalysts. (**e**) Pore size distribution plot of Ni-N-CF and Ni-PVDF catalysts.

**Figure 3 nanomaterials-15-00473-f003:**
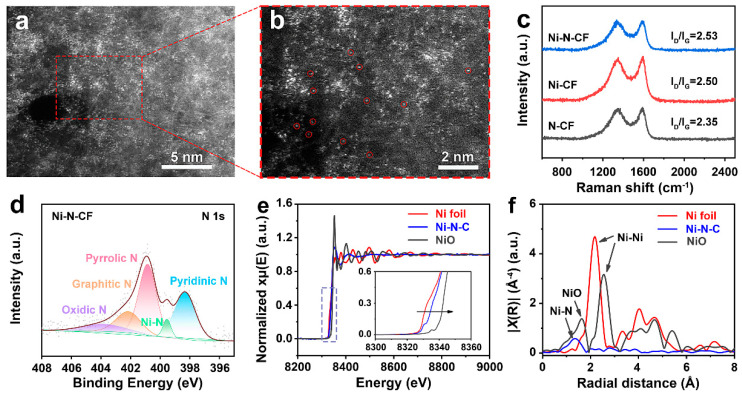
(**a**,**b**) HAADF-STEM images of Ni-N-CF. (**c**) The Raman spectra of Ni-N-CF, Ni-CF, and N-CF. (**d**) N 1s spectra of Ni-N-CF catalysts. (**e**) Ni K-edge XANES of the Ni foil, Ni-N-C, and NiO. (**f**) EXAFS spectra at the Ni K-edge for Ni-N-CF.

**Figure 4 nanomaterials-15-00473-f004:**
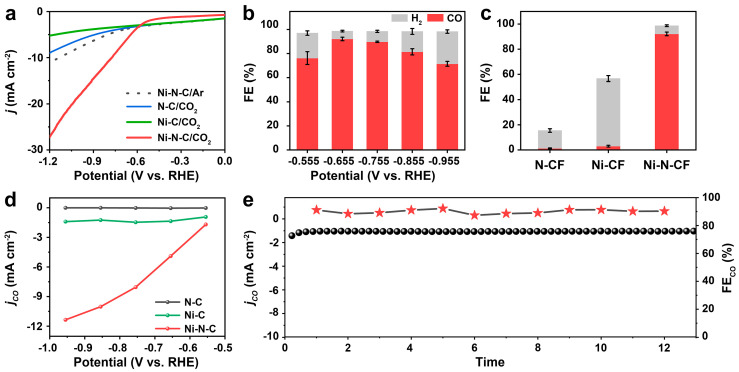
(**a**) LSV curves of Ni-N-CF in Ar-saturated 0.5 M NaHCO_3_ solution and LSV curves in CO_2_ saturated 0.5 M NaHCO_3_ solution of N-CF, Ni-CF and Ni-N-CF. (**b**) Faradaic efficiency of CO product in CO_2_RR on Ni-N-CF at various cathode potentials. (**c**) Faradaic efficiency CO product in CO_2_RR on N-CF, Ni-CF, and Ni-N-CF at −0.65 V vs. RHE. (**d**) The partial current density for CO production of Ni-N-CF, Ni-CF, and N-CF at different voltages. (**e**) Stability test of Ni-N-CF at −0.65 V vs. RHE over 12 h.

## Data Availability

Data are contained within the article and Appendix A.

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
