# Peer review of "Constructing of Ni-Nx Active Sites in Self-Supported Ni Single-Atom Catalysts for Efficient Reduction of CO2 to CO"

_nanomaterials, 2025, doi:10.3390/nano15060473_

Round 1

Reviewer 1 Report

Comments and Suggestions for Authors

The article reported a novel approach for synthesizing self-supported Ni single-atom catalysts (SACs) for efficient electrochemical COâ‚‚ reduction to CO. Using electrospinning, a Ni-doped polyvinylidene fluoride (PVDF) fiber membrane was fabricated and converted into a nitrogen-doped three-dimensional catalyst through controlled carbonization, ensuring structural integrity and stabilization of Ni atoms in an Ni-Nâ‚„ configuration. However, there is some issues that need further clarification. I suggest this article to be reconsider after revisions. Below are detailed comments.

  1. How did the authors verify the Ni-Nâ‚„ structure?
  2. In Figure 4b and c, what is the rest of Faraday efficiency when the sum of H2 and CO is not 100%.
  3. How is the performance of the catalysts reported compared to other literature?
  4. In Figure S3, please translate the label in the figure to English.

Author Response

  Thank you very much for your precious comments and suggestions concerning our manuscript entitled “Constructing of Ni-Nx Active Sites in Self-supported Ni Single-Atom Catalysts for Efficient Reduction of CO2 to CO’’ (original manuscript No.: nanomaterials-3513332). In this response letter, your comments are presented in black italics, our responses are in blue, and all changes are marked in red color in the revised manuscript and supporting information. These comments are all valuable and very helpful for revising and improving our paper. We have studied comments carefully and have made corrections which we hope meet with your approval.For the readability of the charts and graphs, please ask the reviewers to download"reply to reviewer".We sincerely thank you for your review.

Response 1: Many thanks for the reviewer’s precious suggestion. As shown in Fig. 3e, the Ni K-edge XANES spectra show that the near-edge absorption energy of Ni-N-C was located between those of the Ni foil and NiO, indicating that the average valence state of Ni in Ni-N-C was between 0 and +2. The fingerprint peak was around 8336 eV, which was assigned to the 1s - 4pz transition of the square planar Ni-N4 moiety [56].

In the revised manuscript, we have added the corresponding description as follows (The aforementioned references have been included in the revised manuscript (References [56]). This change can be found in the revised manuscript – page 7 and line 253-261.):

“As shown in Figure 3e, the Ni K-edge XANES spectra show that the near-edge absorption energy of Ni-N-C was located between those of the Ni foil and NiO, indicating that the average valence state of Ni in Ni-N-C was between 0 and +2. The fingerprint peak was around 8336 eV, which was assigned to the 1s - 4pz transition of the square planar Ni-N4 moiety [56]. Figure 3f displayed the Fourier transform of the EXAFS spectrum with k³-weighted χ(k) function, where only one peak at 1.39 Å was observed in Ni-N-CF, corresponding to the Ni-N bond.[16] This indicated that Ni atoms were isolated and dispersed on the nitrogen-doped carbon support.”(The corresponding figure here is for the reviewer to download the "reply to reviewer", thank you!)

Comments 2: In Figure 4b and c, what is the rest of Faraday efficiency when the sum of H2 and CO is not 100%.

Response 2: We are deeply grateful for the reviewer’s invaluable suggestions. After adjusting the sealing of the H-type reaction electrolytic cell, we repeated the performance tests of the samples three times and added error bars in Figures 4b and 4c. Considering experimental errors, the total Faradaic efficiency of CO and Hâ‚‚ for Ni-N-CF during COâ‚‚ reduction could reach 100% (Figure 4b). However, due to the losses and lower current density in N-CF and Ni-CF, the total Faradaic efficiency of CO and Hâ‚‚ for these catalysts was less than 100% (Figure 4c).

In the revised manuscript, we added the corresponding description as follows (This change can be found in the revised manuscript – page 8 and line 277-281.):

“The activity and selectivity of the Ni-N-CF catalyst were tested after one hour of electrolysis at a constant potential. Only CO and H2 were detected over the whole investigated potential range. The maximum FE of CO reached 92% at −0.65 V vs RHE (Figure 4b), which the selectivity of Ni-N-CF for reduction to CO was much higher than N-C (1%) and Ni-CF (3%) (Figure 4c).”(The corresponding figure here is for the reviewer to download the "reply to reviewer", thank you!)

Comments 3: How is the performance of the catalysts reported compared to other literature?

Response 3: We appreciate the valuable comments from the reviewers. To verify that our catalyst exhibits better catalytic performance compared to other advanced catalysts, we had reviewed about eight papers and presented a comparison in tabular form with other studies (Table S2, supporting information). We found that the Ni-N4 catalyst still demonstrates significant advantages in terms of Faradaic efficiency (92%).

In the revised article, we added the corresponding description as follows (Content change can be found in the revised article -page 8 and line 290-292. The aforementioned references have been included in the supporting information. (References [1]-[8]). This change can be found in Figure S2, supporting information – page 14 and line 69-97.):

“In addition, compared to other recently reported Ni-Nâ‚„ catalysts, Ni-N-CF seems to exhibited superior selectivity for CO products in COâ‚‚ RR (Table S2, supporting information)”.(The corresponding figure here is for the reviewer to download the "reply to reviewer", thank you!)

Comments 4: In Figure S3, please translate the label in the figure to English.

Response 4: Thanks to the comments. We have checked and revised the wrong caption in Figure S3 in the Supporting information.

In the revised Supporting information, we have modified Figure S3(now is Figure S2, supporting information). This change can be found in Figure S2, supporting information – page 4 and line 28-31.(The corresponding figure here is for the reviewer to download the "reply to reviewer", thank you!)

Reviewer 2 Report

Comments and Suggestions for Authors

The manuscript titled "Constructing Ni-Nx Active Sites in Self-Supported Ni Single-Atom Catalysts for Efficient COâ‚‚ Reduction to CO" presents the preparation of Ni-N-CF-based electrodes for COâ‚‚ reduction to CO. While the material is novel and presents some innovative aspects in the literature, the manuscript is poorly written and lacks proper organization. I do not recommend accepting the manuscript in its current form. However, I suggest resubmission after addressing major revisions.

  1. What is the goal of the fiber membrane? What is its actual geometric area? What advantages does this configuration offer compared to those reported in the literature?
  2. I recommend including the most relevant details in Section 2: Materials and Methods.
  3. What is the state of the art in the electrochemical reduction of COâ‚‚ to CO? What types of catalysts have been previously tested? It is important to emphasize the main novelty of the study.
  4. The results should be reorganized into different subsections.
  5. The Conclusions should be rewritten, detailing point by point the main advances achieved in the manuscript so far.
  6. Compare the results with those achieved in the literature.
  7. What other byproducts have been detected apart from CO and Hâ‚‚?
  8. What is the average size of the catalyst? What is the thickness of the electrode? What layers does the electrode have? What about the counter electrode? There are many unanswered questions.
  9. There are too many figures related to characterization, but I miss a more detailed interpretation of the characterization results.
  10. What phenomena are involved in the electrochemical reduction of COâ‚‚ to CO using a Ni-N-CF-based electrode? This type of material is more commonly used for the oxygen evolution reaction.
  11. What are the achieved concentrations of CO and Hâ‚‚? What are the reaction rates and energy consumption values? Additional figures of merit should be included.
  12. More relevant literature should be included, as this is a hot topic.

Author Response

Thank you very much for your precious comments and suggestions concerning our manuscript entitled “Constructing of Ni-Nx Active Sites in Self-supported Ni Single-Atom Catalysts for Efficient Reduction of CO2 to CO’’ (original manuscript No.: nanomaterials-3513332). In this response letter, your comments are presented in black italics, our responses are in blue, and all changes are marked in red color in the revised manuscript and supporting information. Your meticulous and insightful suggestions have infused new vitality into our paper. Your professional dedication and rigorous approach serve as a model for us to learn from. We extend our heartfelt respect and gratitude to you.For the readability of the charts and graphs, please ask the reviewers to download"reply to reviewer".We sincerely thank you for your review.

Reviewer 3 Report

Comments and Suggestions for Authors

This research focuses on the synthesis, characterization, and electrochemical testing of three types of catalysts: Ni-doped PVDF catalyst carbonized in an argon-ammonia blended atmosphere (Ni-N-C), Ni-doped PVDF catalyst carbonized in an argon environment (Ni-C), and PVDF carbonized under an ammonia atmosphere (N-C). The article clearly presents the background and methodology of the catalyst preparation, highlighting the role of nitrogen and nickel doping in enhancing the catalytic properties.

The study provides an in-depth analysis of the synthesis procedure, which involves the electrospinning of a PVDF solution containing nickel chloride, followed by cross-linking treatments and carbonization under controlled conditions. The detailed description of the experimental setup, including the electrospinning parameters (e.g., needle-to-collector distance, voltage, and flow rate), vacuum drying, and hot pressing conditions, shows a well-structured approach. Furthermore, the carbonization process, particularly the use of an argon-ammonia mixed atmosphere for nitrogen doping, is carefully detailed, ensuring reproducibility and clarity.

The experimental design is systematic, and the characterization techniques are comprehensive, with the preparation steps for each catalyst type clearly defined. The study also includes control samples (N-C and Ni-C) to assess the impact of nickel and nitrogen doping, providing a robust framework for comparing the different catalysts.

Advantages: The experimental design is meticulous, the characterization techniques are well-suited to the research goals, and the methodology is clearly explained. The stepwise approach for catalyst preparation allows for a deep understanding of the process, and the inclusion of control samples enhances the study's validity.

Deficiencies: While the methodology is well-detailed, there is insufficient discussion on the mechanisms that govern the catalytic performance of the materials, especially in relation to the roles of nickel and nitrogen doping. The innovative aspects of the study, such as the impact of nitrogen doping on the catalyst's electrochemical properties, could be emphasized more clearly. Additionally, there is a lack of comparison between the prepared catalysts and other similar catalysts in the literature, which would provide a clearer context for evaluating the catalysts' performance.

Author Response

Thank you very much for your precious comments and suggestions concerning our manuscript entitled “Constructing of Ni-Nx Active Sites in Self-supported Ni Single-Atom Catalysts for Efficient Reduction of CO2 to CO’’ (original manuscript No.: nanomaterials-3513332). In this response letter, your comments are presented in black italics, our responses are in blue, and all changes are marked in red color in the revised manuscript and supporting information. We sincerely appreciate your affirmation of our manuscript, which has been both a tremendous encouragement and a strong impetus for our future research endeavors. Your valuable suggestions have further served as a guiding beacon in refining the articulation of the paper's innovation. Please allow us to express our deepest appreciation for the cost of your time and the effort you put in.For the readability of the charts and graphs, please ask the reviewers to download"reply to reviewer".We sincerely thank you for your review.

Round 2

Reviewer 1 Report

Comments and Suggestions for Authors

Thank you for the response.

Reviewer 2 Report

Comments and Suggestions for Authors

After addressing the reviewer's comments, I recommended its publication in Nanomaterials journal.